# Efficacy of Magnetic Therapy in Pain Reduction in Patients with Chronic Pelvic Pain: A Systematic Review

**DOI:** 10.3390/ijerph19105824

**Published:** 2022-05-10

**Authors:** Alicia María de Pedro Negri, María Jesús Ruiz Prieto, Esther Díaz-Mohedo, Rocío Martín-Valero

**Affiliations:** Department of Physiotherapy, Faculty of Health Science, Ampliacion de Campus de Teatinos, University of Malaga, C/Arquitecto Francisco Peñalosa 3, 29071 Malaga, Spain; aliciadepedronegri@gmail.com (A.M.d.P.N.); marsuruiz18@gmail.com (M.J.R.P.); estherdiaz@uma.es (E.D.-M.)

**Keywords:** magnetic field therapy, chronic pelvic pain, pain

## Abstract

Chronic pelvic pain (CPP), also known as chronic pelvic pain syndrome (CPPS), is a common and painful condition. However, its treatment is still a challenge. The findings about the beneficial effects of electromagnetic therapy provide a new, potentially valid, therapeutic alternative for the management of patients with CPP. Objectives: to analyze the efficacy of magnetic field therapy in pain reduction in patients with CPP and for other variables, such as urinary symptoms and quality of life, as well as to review the evidence, in order to establish an action protocol. A qualitative systematic review was carried out, based on the PRISMA protocol and registered in PROSPERO (CRD42022285428). A search was performed in the PubMed, Medline, Scopus, Cochrane, PEDro, BVS, and WOS databases, including those articles in which the patients suffered from CPP; the study variable was pain, and the intervention was based on the application of magnetic fields. Results: Among the 81 articles found, five clinical trials were considered (with an average score of 7.2 in the PEDro scale), with a total of 278 participants, most of whom presented improvements in perceived pain (*p* ≤ 0.05), as well as in quality of life (*p* < 0.05) and urinary symptoms (*p* = 0.05), evaluated through the NIH-CPSI and VAS scales. The therapy was conducted as a monotherapy or in combination with a pharmacological treatment. There was no common protocol among the different articles. Conclusions: Intervention programs through electromagnetic therapy, on their own or with other therapies, can be effective in patients with CPP.

## 1. Introduction

Chronic pelvic pain (CPP) is a complex clinical condition that has been defined as a persistent pain perceived for at least 6 months in structures related to the pelvis, associated with negative cognitive, behavioral, sexual and emotional alterations, and with a clear impact on the quality of life of people who suffer from this disease [1,2]. Currently, the terms of pain syndromes have been introduced to indicate the various mechanisms involved, both physical and psychological, so that the concept of chronic pelvic pain syndrome (CPPS) was developed, called “pain as a disease process”. If the pain is poorly localized or is perceived in three or more foci, it is diagnosed as chronic pelvic pain syndrome (CPPS), without the need for further subdivision by effector systems or organs. However, the perception of pain can be focused within a single organ, more than one pelvic organ, or even associated with systemic symptoms, so that the term chronic primary pelvic pain syndrome (CPPPS) was introduced to refer to this type of nonspecific pain, poorly localized and without obvious pathology. These subdivisions of chronic pelvic pain should only be used if there is adequate evidence to support their use, which is why the general approach has been taken to refer to all perceived pelvic pain as CPPS [3].

It is an important problem, due to both its frequency and morbidity, with a prevalence of 12% and a rate of 33% throughout life, which is similar to the prevalence of disorders such as asthma and low back pain. CPP/CPPS affects both men and women, although the rate is higher among females [1,4].

Diagnosing and managing chronic pelvic pain is difficult and confusing, as a result of the many different and frequently divergent symptoms across several body systems. Many interpretations have been described about the multi-symptoms present in chronic pelvic pain, with four possible explanations for the confusing presentation of chronic pelvic pain: viscerovisceral convergence, viscerosomatic convergence, hypertonicity of pelvic floor musculature, and central sensitization [5,6]

It presents a varied etiology that comprises gynecological, gastrointestinal, and musculoskeletal causes [5], with the four most frequent etiological diagnoses being: endometriosis [6], inflammatory pelvic disease adhesions, irritable bowel syndrome, and interstitial cystitis [1,7].

All these aspects, not only hinder the diagnosis and subsequent therapeutic approach to CPP, but they also lead to considering it as a multidisciplinary and multifactor clinical entity [5]. Therefore, in order to carry out a correct diagnostic approximation, it is necessary to perform a detailed clinical history, a proper physical examination with bimanual palpation, which allows identifying different structures causing painful conditions; as well as the use of laboratory tests indicated to discard chronic inflammatory processes; imaging tests such as ultrasound to detect anatomical alterations; and surgical tools such as laparoscopy, which is very useful when there is strong suspicion of endometriosis [4,6,8,9].

The therapeutic approach is based both on the treatment of CPP itself and on the treatment of diseases and disorders that may cause or contribute to CPP. The treatments can be classified as pharmacological, psychological, surgical, or physiotherapeutic. Within the physiotherapeutic approach, there is a wide range of treatments: sacral and pudendal neuromodulation, laser therapy, manual therapy, and therapeutic exercise, among others [10].

However, in the last decade, other treatment modalities have been described, such as electromagnetic therapy, which poses a novel approach, presenting the same underlying effect as electric stimulation, due to the interaction with the nervous system, although, in this case, the electromagnetic field goes non-invasively through the neuromuscular tissue, where the induced electric currents depolarize the neural cells, thus altering the resting membrane potential and, thereby, reducing the transmission of painful impulses [11,12].

Magnetic stimulation of the pelvis produces a direct stimulus in muscular trophism, favoring an anti-inflammatory effect, in addition to its relaxing and de-contracting effect, as it reduces the sympathetic tone; thus, restoring the normal muscular activity of the pelvic floor [11,13].

This therapy poses a novel and promising approach for the treatment of CPP/CPPS, given its multiple benefits; where low intensities (10–20 Hz) are associated with analgesic and myorelaxant effects, whereas medium-high intensities (≥50 Hz) are more strongly related to an anti-inflammatory and tissue-repairing effect [11,14,15].

The aim of this systematic review was to analyze the efficacy of magnetic field therapy in pain reduction in men and women with CPP, as well as to determine the characteristics of the protocols of magnetic fields applied in the studies, the variables related to pain, the most popular measurement tools, and the efficacy of magnetic field therapy with respect to other variables, such as quality of life. The hypothesis was that magnetic field therapy would result in pain reduction and improved quality of life in men and women with CPP.

## 2. Materials and Methods

### 2.1. Study Design

A systematic review was conducted following the “preferred reporting items for systematic review and meta-analyses” (PRISMA) recommendations [16]. The PRISMA checklist is detailed in Appendix A. The review protocol was registered in the PROSPERO international registry database [17] (CRD42022285428). All analyses were carried out with data from previously published studies; therefore, neither ethical approval nor consent from the patients were required.

A research question was established based on the PICO strategy [18], with the following inclusion criteria: (I) men and women with chronic pelvic pain; (II) intervention with high-intensity magnetic fields; (III) induced pain as one of the variables; (IV) randomized controlled trials and pilot studies.

### 2.2. Search Strategy

A literature search was performed in 6 databases: PubMed, Scopus, Cochrane, PEDro, Web of Science (WOS), and Medline, and in the search engine Virtual Health Library (BVS). The search was focused on randomized controlled trials, identifying clinical studies that addressed the effect of magnetic fields in patients with CPP/CPPS.

The recovery and selection of articles was carried out by two independent researchers (ADP and MJR), using the following combinations of keywords: “magnetic field therapy” and/or “electromagnetic stimulation” and “chronic pelvic pain” or “chronic pelvic pain syndrome”. The search strategy is detailed in Table 1. For inclusion of the studies, we used the methodological quality evaluation scale of the PEDro database, [16] and the inclusion and exclusion criteria of the review. Another two researchers (EDM and RMV) were in charge of making decisions when the first two researchers did not reach a consensus.

### 2.3. Inclusion and Exclusion Criteria

The review included all randomized controlled trials and pilot studies in which the subjects were diagnosed with chronic pelvic pain, where at least one of the two groups used magnetic field therapy as the intervention, and pain was analyzed as a variable.

On the other hand, the review excluded those articles that did not respond to the research question established based on the PICO model [18]; that is, all those studies which were not RCTs or pilot studies with magnetic field interventions vs. placebo, studies with no treatment or other interventions, those whose intervention group did not present CPP, and, lastly, those which did not report complete results.

### 2.4. Eligibility Criteria and Internal Validity Assessment

All the studies identified in the search were evaluated by three reviewers (ADP, MJR, and RMV), to decide their inclusion. Initially, after the exclusion of duplicate studies, the articles were selected based on the title and abstract. After reading a complete text and considering it adequate, its relevance was assessed according to the inclusion and exclusion criteria. For the eligible trials, once all three reviewers had completed the data extraction independently, a fourth reviewer cross-verified their consistency. Three reviewers (ADP, MJR, and RMV) assessed the methodological quality. In case of doubt, authors resolved disagreements by consensus, and by consulting a fourth author (EDM) when necessary.

The review included clinical trials that described the obtained results related to pain in patients with CPP. No limitations were established with respect to language, publication date, or patients’ age or sex. This review required that the study results measured the effects of magnetic field therapy on the symptoms of the patients with CPP. There were no limitations regarding the comparison of interventions or in the magnetic field treatment used.

The quality of the scientific studies was determined using the PEDro scale [19], which is a methodological quality scale for controlled trials. The scale consists of 11 items, of which only items 2–11 have a score, since the first item (“selection criteria”) is not used to calculate the score, as it affects the external validity. Articles obtain a score of 0 to 10 from the methodological perspective, as follows: (I) 9–10: “excellent”; (II) 6–8: “good”; (III) 4 or 5: ”acceptable”; (IV) <4: “deficient”. A study with a PEDro score of 6 or higher is considered level-1 evidence, whereas those with a score of 5 or lower are considered low-level evidence.

## 3. Results

### 3.1. Study Selection and Methodological Quality Evaluation

The initial search in the different databases produced 81 studies that were potentially relevant. In the search strategy, the results were filtered by randomized controlled trial (RCT). After obtaining the articles from the seven databases, the duplicate results were reviewed and removed; thus, reducing the sample to 48 articles. Then, the titles and abstracts of each of the articles were analyzed to ensure their eligibility. Lastly, the inclusion/exclusion criteria were applied to them, reducing the final sample of this qualitative review to five articles. A quantitative systematic review or meta-analysis could not be conducted, since it was not possible to statistically combine the results of the studies. The main steps related to the search process are represented in Figure 1.

In the evaluation of the articles, according to the PEDro scale, the values ranged between 4 and 9. Table 2 and Table 3 show the methodological quality evaluation, with an average score of 7.2 out of 10 points. We found one article with excellent methodological quality [20], three with good quality [21,22,23], and one with poor quality [24]. In all articles, the selection criteria were specified, and they reported the comparison of the results of the variable “pain” between groups. Regarding criterion 6, most studies lacked blinding of the therapists [20,21,23,24], since they knew, at all times, the treatments applied to the different groups. Given that only five articles were included in this qualitative synthesis, it was considered that the methodological quality of the trials was sufficient if their score was at least 4 out of 10 points.

### 3.2. Description of the General Characteristics of the Studies Included

This systematic review included five studies, with a total of 278 patients clinically diagnosed with CPP/CPPS. The impact of pain was evaluated using the National Institute of Health Chronic Prostatitis Symptom Index (NIH-CPSI) in most studies [17,18,20], with the aim of quantifying the response to the treatment. The age of the participants ranged between 18 and 67 years. Table 4 and Table 5 show the results related to the characteristics of the studies and of the interventions, as well as the rest of the information extracted from the articles.

Although no limitation was established with respect to the year of publication, the trials included in the synthesis are relatively recent, with two of them being published in the last seven years [17,18]. With respect to the therapeutic interventions, despite the fact that all the studies carried out an intervention based on the application of magnetic fields, two of them used an electromagnetic chair as the therapeutic method [20,21], one applied sono-electro-magnetic therapy [17], another one used bipolar magnets [19], and the most recent study, i.e., that of He et al., 2020, used high-intensity magnetic electrostimulation as the intervention method [18].

### 3.3. Description of the Characteristics of the Study Sample

Differences were observed in the way in which CPP was defined in terms of pain duration, with two of the articles describing it as a pain that persists for at least 3 months [17,18], whereas the three remaining studies did not consider CPP in an interval shorter than 6 months [19,20,21].

The sample size was fairly homogenous among the studies, with all of them using small samples that ranged between 21 and 124 participants. Moreover, they presented similarities in terms of sex, age, and level of severity [17,18,19,20,21]. It is worth highlighting that, in most of the trials [17,18,20,21], the study population constituted men with CPP/CPPS; therefore, of the total of 278 patients, 245 were men (88.13%) and 33 were women (11.87%).

With respect to the inclusion criteria, all the trials required that the intervention participants were over 18 years of age, with no evidence of infection or other pathologies and with pain sensation in the pelvic and abdominal structures [17,18,19,20,21]. One of the articles also required the intervention participants to have a total score of NIH-CPSI ≥15 and a pain sub-score of NIH-CPSI ≥8 [17].

Regarding the exclusion criteria, four articles [17,18,20,21] discarded those individuals with a documented history of bladder or prostate cancer, urethral surgery, or previous pelvic radiotherapy. In the study with a female intervention population [19], the exclusion criteria included pregnancy or lactation period and any unstable medical disorder that was not controlled by the standard treatment.

### 3.4. Description of the Intervention Group and Control Group

Three of the studies conducted an isolated intervention of electromagnetic stimulation [17,19,21], whereas two of them combined the electromagnetic therapy with a pharmacological treatment [18,20]. With regard to the control group of those trials that performed the treatment as a monotherapy [17,19,21], a placebo therapy was applied; that is, the intervention was identical, but the magnetic device was deactivated. On the other hand, in the articles that combined the magnetic therapy with medication [18,20], the same drug was administered to both the experimental and control groups.

#### 3.4.1. Description of the Intervention Protocol of the Electromagnetic Therapy

With regard to the intervention protocols, the analyzed studies present different characteristics in terms of performance within the treatment sessions. The most notable differences lie in the magnetic field pulse frequencies and intensities used, the treatment or session duration, and the number of sessions [17,18,19,20,21].

#### 3.4.2. Description of the Intervention Protocol of the Comparison Group

The studies can be grouped based on the execution of the intervention through active therapy or placebo, [17,19,21], and the groups in which an active treatment was applied, which, in this case, consisted in the administration of drugs [18,20].

In addition, there was no uniform protocol in terms of drug dosage and therapy duration in those trials that administered drugs to the control group [18,20].

### 3.5. Description of the Variables and Measurement Tools

The primary variable recorded in all studies was pain [17,18,19,20,21]; evaluating pain intensity with present pain intensity (PPI), visual analogic scale (VAS) and/or National Institute of Health Prostatitis Syndrome Index (NIH-CPSI), pain disability index (PDI), and McFill Pain Questionnaire (MPQ). Statistically significant improvements were observed in this variable (*p* ≤ 0.05) in four of the studies included in this review [18,19,20,21].

As secondary variables, all five studies measured the urinary symptoms and the quality of life [17,18,19,20,21]. Greater differences were found between the groups for the score of quality of life, which was measured with NIH-CPSI; [17,18]

VAS [21], which is a simple pain measurement method, in which, from a 10 cm line, the continuous spectrum of the painful experience is represented, expressing “no pain” at one end and “the worst pain imaginable” at the other end (Cronbach’s alpha: 0.94) [22]. NIH-CPSI was used in three studies [17,18,20]. This index evaluates the three major domains of PC/DPC; that is, pain, micturitional alterations, and impact on the quality of life (Cronbach’s alpha: 0.864) [23].

PPI scale was used in one of the studies to measure the magnitude of the perceived pain [19]. This presents a high degree of reliability (Cronbach’s alpha: 0.74) [24]. PDI evaluates the degree of pain perceived by the patient and its interference in seven areas of daily activity (Cronbach’s alpha: 0.912) [25].

MPQ, provides a reliable valuation ([: 0.93) [26] of pain from a triple perspective: sensory, affective-emotional, and evaluative [26]. Clinical global impression (CGI) was also used [19] to quantify and carry out a follow-up of the patient’s response to the treatment during the intervention in a safe manner (Cronbach’s alpha: 0.86) [27].

Other articles [20] included the international prostate symptoms score (IPSS), which provides an objective estimation of the intensity of the symptoms of the lower urinary tract of the patient. It is a useful and highly reliable tool (Cronbach’s alpha: 0.72) [28] in the evaluation of the need for pharmacological treatment and the patient’s response to it [20].

## 4. Discussion

A systematic review was carried out to synthesize the scientific evidence and evaluate its quality regarding the efficacy of electromagnetic therapy in patients who presented with chronic pelvic pain. The obtained results support the idea that magnetic field therapy is effective in reducing pain in patients with CPP. Regarding the of level of evidence, four studies found level 1 and the other article level 2. However, the analysis of the literature related to the intervention design must be interpreted with caution, due to the differences in the application of the treatment technique, the intervention time, and the number of sessions.

### 4.1. Characteristics of the Sample

With regard to the characteristics of the sample, it was heterogeneous samples regarding gender, as the number of studies in men was higher than in women. The intervention was applied in women in only one of the studies [19], which makes it difficult to extrapolate the results to the female population with CPP; consequently, it would be useful to improve or consider this aspect in future research, recruiting more homogeneous samples regarding gender.

### 4.2. Measuring Instruments

Concerning measuring instruments, the most commonly used in our paper was the NHI-CPSI scale. Four studies included in this systematic review [20,21,22,23] analyzed the changes using the NHI-CPSI scale [20,21,23]. It has been observed that electromagnetic therapy increases the quality of life of the patients in comparison with other therapies. Specifically, the studies of Kessler et al., 2014 and Paick et al., 2006 showed a greater difference between the intervention and control groups in the score of the quality of life (*p* = 0.015 and *p* = 0.022, respectively) [20,23].

Four studies [20,21,23,24] considered urinary symptoms, mainly the post-micturitional residue, as the outcome variable. The treatments based on magnetic fields reported favorable results in this respect in three of the trials [21,23,24]. This is in line with the results obtained by Samuels et al., 2019, in a prospective study conducted in women with urinary incontinence, who received high-intensity magnetic electrostimulation at the level of the perineum, obtaining a significant improvement in urinary symptoms and a better control of micturition [13].

Only one of the studies considered the sexual behavior of the participants [22]. The score of this variable in the intervention group was substantially better after the end of the magnetic therapy (*p* = 0.02). The same was observed in the study of Samuels et al., 2019, who obtained satisfactory results regarding the muscular tone of the pelvic floor, sexual desire, and orgasm intensity [13].

### 4.3. Intervention Strategies of Magnetic Field Therapy

Among the studies included in this review, three of them carried out a magnetic treatment as monotherapy [20,22,24], whereas the other two trials combined magnetic electrostimulation with a pharmacological treatment [21,23]. Four of the studies showed significant improvements in pain reduction in the medium-to-long term [21,22,23,24]; in turn, the studies that combined a magnetic therapy with medication presented promising results in a shorter interval, at 2–6 weeks [21,23]. In the same line, the trials of He et al., 2020 and Paick et al., 2006 showed better results in the control group with respect to the non-experimental groups of the rest of the studies, where no intervention was applied [20,22,24].

Regarding the magnetic field dosage, two of the studies report an association between the use of a pulse frequency of 10 Hz and the attainment of an analgesic effect on the nerve endings [23,24]. Furthermore, all five trials related higher frequencies (pulse frequency of 50 Hz and pulse intensity of 500 G) to an antiedematous and stimulating effect of tissue repair [20,21,22,23,24]. Previous studies have observed similar effects, in which pulse frequencies of 1–50 Hz showed a significant increase of local blood flow in the stimulated area and of collagen synthesis; thus, improving the state of the ischemic tissue [14,15].

The treatment sessions varied among the studies, from one session per day [20,21,22] to two sessions per week [23,24], with a duration of 2–16 weeks. The duration of the sessions varied between 10 min [20] and 24 h [22]. Some studies showed that 20–30 min of electromagnetic stimulus generated satisfactory results with respect to the reduction of pain perception [13,24], which is in line with the findings presented in the studies by Rowe et al., 2004 and Paick et al., 2006, where sessions of 25 min were sufficient to provoke a significant decrease in pain sensation (*p* < 0.05).

### 4.4. Implications for Clinical Practice

It can be asserted that larger sample sizes are required, as well as standard interpretation models to measure the variables, and standardized data to validate the existing literature and to, thus, be able to attain an optimal electromagnetic field treatment protocol.

Correct evaluation of the efficacy of magnetic stimulation for pain reduction requires measuring and calculating a variety of parameters, such as amplitude, field gradients, and exposure duration. Furthermore, not only the precise characteristics of the magnetic field applied must be taken into account, but also the exact diagnosis and all the relevant clinical data. Additional research on magnetic stimulation should identify which magnetic fields can be detected by cells or subcellular structures, and determine the cellular and tissue responses to the applied signals. These evaluations are important, since there is an increasing number of magnetic and electromagnetic technologies and devices that are used in clinical practice.

However, the good results reported in the five trials, the scarcity of adverse effects, the low rate of abandonment, and the non-invasive nature of the electromagnetic treatment lead us to consider its incorporation into clinical practice. This would reduce the excessive intake of drugs, which, despite being effective in reducing the symptoms of patients with CPP [18,20], can also block the neurotransmitters and neuroreceptors that modulate pain; thus, perpetuating and even worsening the symptoms [29,30,31].

### 4.5. Limitations

There are several limitations to the analysis that should be considered when interpreting these results. First, the heterogeneity among the different studies was so extensive that a meta-analysis could not be performed. It was difficult to find articles on the study topic, despite the search conducted in multiple electronic databases of published studies. Second, there was little uniformity in study populations, sample sizes, interventions and their duration, measured variables, and measurement instruments. All the articles presented in this review were quite heterogeneous in certain parameters; moreover, one of them presented a moderate methodological quality [21]. Therefore, the results must be interpreted with caution, since the studies included in the review used different devices, treatment times, and times between sessions.

Another possible source of bias is the lack of information about the drug used in one of the articles [18], which does not reveal its composition, pharmacokinetics, or pharmacodynamics, and does not provide enough information for correct administration and use of the drug; therefore, it is not possible to safely certify the efficacy of the clinical indications to which it is targeted.

Another reason that leads to consider the possible effects of biases and to be cautious in the interpretation of the results is the influence of funding on the studies, especially in one of the articles [17], where the source of funding coincided with the brand of the device used for the intervention. Similarly, it is worth highlighting the study of Brown et al., 2002, where compliance with the therapy was determined by self-report [19]; thus, the information provided may be incorrect due to a lack of objectivity.

### 4.6. Future Studies

Further research is required to consider the effects of the different electromagnetic treatment protocols, as well as high-quality clinical trials, with larger sample sizes and a greater homogeneity in terms of the sex of the participants, since only one of the studies analyzed in the review included women [19]. Similarly, it is necessary to unify criteria about the measurement of pain and establish intervention protocols considering the devices used, the frequency and intensity, and the treatment duration.

Moreover, researchers are encouraged to conduct clinical trials that demonstrate the efficacy of this therapy in combination with physical exercise, in order to provide safer therapeutic options; since it is a healthy practice, which has reported favorable results in this profile of patients, despite the scarce literature.

## 5. Conclusions

The intervention programs based on magnetic field therapy, on their own or combined with other therapies, improved the painful symptoms in patients with CPP/CPPS. The results suggest that interventions with magnetic electrostimulation aimed at reducing pain must be included in the treatment programs of patients with CPP.

A great advantage of this therapy is that it is non-invasive, and it has benefits for the other symptoms related to CPP, especially the quality of life and urinary symptoms.

Given the diversity of protocols of the analyzed studies, solid conclusions cannot be drawn to establish the most recommended parameters. However, there is greater inclination for a treatment dosage of twice per week, linked to the better response to the therapy. Regarding the criteria of how to measure the “pain” variable, most of the studies selected the NIH-CPSI scale, thus this can be considered a reliable tool for the measurement of pain in future studies.

## Figures and Tables

**Figure 1 ijerph-19-05824-f001:**
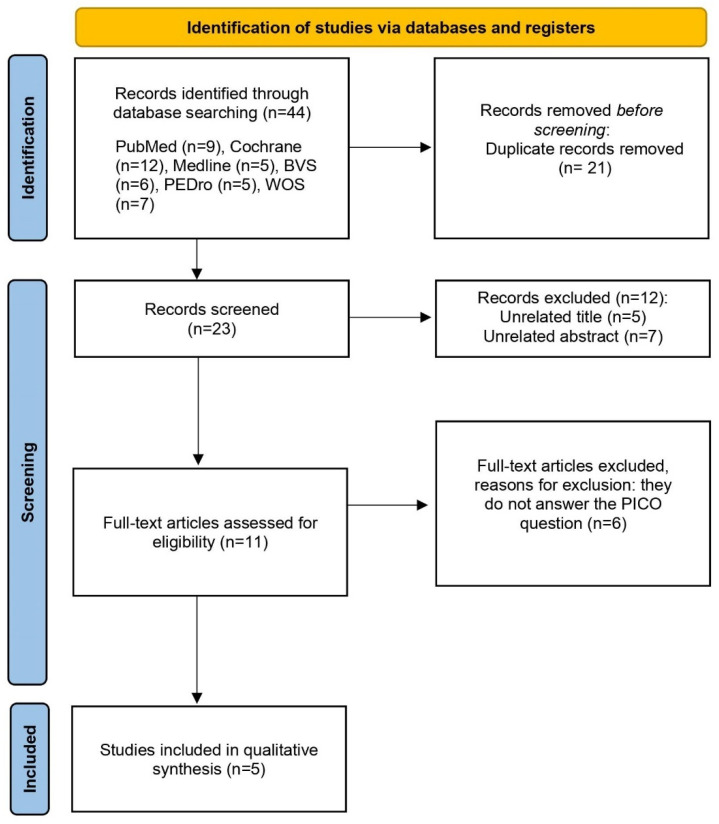
PRISMA flow diagram of the study selection process.

**Table 1 ijerph-19-05824-t001:** Search strategy.

Databases	Search Strategy
PubMed	(“Chronic pelvic pain” OR “chronic pelvic pain syndrome”) AND (“Magnetic stimulation” OR “Electromagnetic” OR “Electromagnetic stimulation” OR “Magnetic neuromodulation” OR “Electro-magnetic therapy” OR “Pelvic electromagnetic therapy”) 9 results
COCHRANE	#1 MeSH descriptor: [Pelvic Pain] explode all trees 1256 #2 MeSH descriptor: [Chronic Pain] explode all trees 2815 #3 MeSH descriptor: [Magnetic Field Therapy] explode all trees 1689 #4 (electromagnetic field therapy): ti, ab, kw 514 #5 (magnetic field therapy): ti, ab, kw 953 #6 MeSH descriptor: [Pain] explode all trees 52801 #7 (pelvic chronic pain): ti, ab, kw 1269 #8 (pelvic chronic pain syndrome): ti, ab, kw 603 #9 (chronic pelvic pain): ti, ab, kw 1269 #10 (chronic pelvic pain syndrome): ti, ab, kw 603 #11 (electromagnetic therapy): ti, ab, kw 872 #12 (magnetic field therapy): ti, ab, kw 953 (#6 AND (#1 AND #2) AND #3) OR ((#7 OR #8 OR #9 OR #10) AND (#4 OR #5 OR #11 OR #12)) 13 results.
BVS	(tw: ((“pelvic chronic pain” OR “chronic pelvic pain” OR “chronic pelvic pain syndrome”))) AND (tw: ((“electromagnetico therapy” OR “magnetic field therapy” OR “extracorporeal magnetic innervation”))) and (tw: (random *)) 6 results
PEDro	SIMPLE: “Magnetic field therapy” “chronic pelvic pain” 1 resultado “electromagnetic therapy” “chronic pelvic pain” 1 resultado “magnetic therapy” “chronic pelvic pain” 2 resultados 4 results ADVANCED: Abstract&title: “magnetic field therapy” “chronic pelvic pain” Therapy: electrotherapies, heat, cold Problem: pain Body part: perineum or genito-urinary system Subdiscipline: continence and women’s health Topic: chronic pain Method: clinical trial. 1 result.
WEB OF SCIENCE	(“Chronic pelvic pain” OR “chronic pelvic pain syndrome”) AND (“Magnetic stimulation” OR “Electromagnetic” OR “Electromagnetic stimulation” OR “Magnetic neuromodulation” OR “Electro-magnetic therapy” OR “Pelvic electromagnetic therapy”) AND (“RCT” OR “Randomized controlled trial”) 7 results.
MEDLINE	(“Chronic pelvic pain” OR “chronic pelvic pain syndrome”) AND (“Magnetic stimulation” OR “Electromagnetic” OR “Electromagnetic stimulation” OR “Magnetic neuromodulation” OR “Electro-magnetic therapy” OR “Pelvic electromagnetic therapy”) 5 results.
SCOPUS	(“Chronic pelvic pain” OR “chronic pelvic pain syndrome”) AND (“Magnetic stimulation” OR “Electromagnetic” OR “Electromagnetic stimulation” OR “Magnetic neuromodulation” OR “Electro-magnetic therapy” OR “Pelvic electromagnetic therapy”) 36 results.

* search engine returns terms that have the same root.

**Table 2 ijerph-19-05824-t002:** PEDro score for methodological quality assessment of five studies.

Study	Total Score	1	2	3	4	5	6	7	8	9	10	11
Brown et al., 2002 [25]	8/10	–	×		×	×	×	×	×		×	×
Kessler et al., 2001 [17]	9/10	–	×	×	×	×	×		×	×	×	×
Rowe et al. [21]	4/10	–	×		×	×						×
Paick et al., 2006 [20]	7/10	–	×	×	×				×	×	×	×
He et al., 2020 [18]	/10	–	×		×	×	×		×	×	×	×

The symbol “×” indicates those items that have been scored; the symbol “–” indicates those items that were not counted for the final score.

**Table 3 ijerph-19-05824-t003:** Benefits of magnetic therapy, based on the level of evidence and grade of recommendation in CPP.

Article	PEDro Puntuation	Article Quality	Level of Evidence
Brown et al., 2002 [19]	8/10	Good	Level 1
Kessler et al., 2014 [17]	9/10	Excelent	Level 1
Rowe et al., 2004 [21]	4/10	Moderate	Level 2
Paick et al., 2006 [20]	7/10	Good	Level 1
He et al., 2020 [18]	8/10	Good	Level 1

**Table 4 ijerph-19-05824-t004:** Revised studies on magnetic therapy in patients with CPP.

Author	Type of Article	Inclusion Criteria	Exclusion Criteria	Intervention Group	Control Group
Rowe E. et al., 2004 [21]	RCT. N = 21 men	<70 years, diagnosed of CPP.	History of cancer Prostate.	Electromagnetic therapy 2 sessions per week for 4 weeks on the chair, Neotonus™. 15 min at 10 Hz and 50 Hz for another 15 min	It was applied with an identical form of device, but without activating it.
Paick J-S et al., 2006 [20]	Pilot study. N = 40 men	Diagnosis of CPP, > 18 years old.	<6 months of symptoms, chronic/acute urethritis, stones in kidney, bladder cancer or prostate, medication antimicrobial/anti-inflammatory 4 weeks before the study.	Terazosin 2 mg once a day for the first 7 days, then, Terazosin 4 mg daily for the next 5 weeks.	Identical pharmacological treatment than the other group and 20 min sessions electromagnetic current in a seat. It was applied twice a week for 6 weeks. 10 Hz during 10 min, and then 50 Hz another 10 min. The system was used Neocontrol, Neonotus Inc.
Kessler TM et al., 2014 [17]	RCT. N = 60 men	Patients with no improvement with other therapies in the last 15 weeks.	Chronic prostatitis bacterial tract infection urinary, post urine residue more than 100 mL, cancer of prostate or urethra and minors of 18 years.	The Sonodyn device device twice a day.	The same treatment was applied but the machine was not activated.
Brown CS et al., 2002 [19]	Pilot study. N = 33 women	18–50 years old, trigger point in abdomen to the palpation, with medication for 6 months, pelvic exam, and normal physique.	Pregnancy, breastfeeding, has a pacemaker or device electronic, having used magnetic therapy previously.	Medical magnets concentric bipolar BIOflex (Avon, Conn) applied at an intensity of 500 G in the surface with a diameter of 50 mm and 1.5 mm wide. Applied 24 h a day for 4 weeks.	The same treatment was applied but the machine was not activated.
He W. et al., 2020 [18]	RCT. N = 124 men	18–50 years old, CPP, not having been treated in the last week	Acute prostatitis and/or chronic bacterial infection, STD, disease metal, stenosis and/or surgery urethral.	Qianlie Beixi Capsules (QBC) 2.4 g was given for 14 days to patients before meals, combined with electromagnetic therapy with the RH device-CZCD Zhengzhou Renhui Medical Equipment Co. It was applied for 30 min a day using 180 mT intensity with 50 Hz.	Qianlie Beixi Capsules for 14 days.

CPP: Chronic Pelvic Pain. NIH-CPSI: National Institute of Health Prostatitis Syndrome Index. STD: Sexually Transmitted Diseases.

**Table 5 ijerph-19-05824-t005:** Key findings of primary studies.

Author	Pain Results	Effects on Other Variables	Desertion
Rowe E et al., 2004 [21]	There was a statistically significant decrease in the intervention group, from 21.7/50 to 14.7/50 at 3 months in VAS; *p* < 0.05). There were no improvements in the placebo group, (*p* > 0.05).	There were no statistically significative improvements of urinary symptoms in the intervention group. There were no improvements in the control.	There were 4 dropouts; 1 in the intervention group, and 3 in the placebo group.
Paick J-S et al., 2006 [20]	There was a statiscally significant improvement in NIH-CPSI score of the group with electromagnetism (*p* < 0.041). Intervention group had statistically significant improvements compared to the pharmacological group (*p* = 0.007).	PVR values and peak flow did not change after treatment. There were improvements in QOL in both groups, in the intervention group from 9.5 to 6.5 (*p* = 0.005), and in the case of the control group from 8 to 6 (*p* = 0.003). I-PSS obtained a better result in elegtromagnetism group with electromagnetism (*p* = 0.002) than in the pharmacological group (*p* = 0.55)	-
Kessler TM et al., 2014 [17]	There was a decrease in NIH-CPSI scale in the intervention group (*p* = 0.7) and in the control group (*p* = 0.24).	Regarding quality of life, there were statistically significant differences between both groups (*p* = 0.015) Both groups had improvements in urinary symptons, being higher in intervention group, although they were not statistically significant (*p* > 0.05). There were no differences in the rest of the variables.	There was 1 dropout.
Brown CS et al., 2002 [19]	The MPQ score showed great improvements in the intervention group patients, but these were not statiscally significant. PPI was reduced, but this was not statistically significant (*p* > 0.05). The PRI-T score improved in the intervention group (*p* > 0.08). The PDI score improved significantly in the intervention group compared to 4% of placebo group (*p* < 0.02).	In the CGI-S score, symptoms improved significant in 28% (*p* < 0.007) compared to the group placebo that improved by 10% (*p* < 0.02).	The study was divided into two parts, one part that lasted 2 weeks with 1 dropout; and another lasted 4 weeks, and there were 13 dropouts.
He W et al., 2020 [18]	There was a statistically significant decrease in NIH-CPSI scale (*p* < 0.05).	Regarding micturition symptoms, the control group improved from the initial, being statistically significant (*p* < 0.05); however, for pain, the intervention group had statistically significant improvements compared to pretreatment and the control group. In QoL, in both groups there were statistically significant improvements.	There was not any desertion.

VAS: visual analogic scale. NIH-CPSI: National Institute of Health Porstatitis Syndrome Index. QOL: quality of life. IP-SS: International Prostate Symptom Score. PVR: post void residual. RPM: residuo post miccional. PSA: antígeno prostático específico. PPI: present pain intensity. PRI-T: quality of pain. PDI: pain disability index. CGI: clinical global impressions scale.

## Data Availability

Not applicable.

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
