# Peer review of "Efficacy of Magnetic Therapy in Pain Reduction in Patients with Chronic Pelvic Pain: A Systematic Review"

_ijerph, 2022, doi:10.3390/ijerph19105824_

Round 1

Reviewer 1 Report

Abstract/Introduction:

  • P1, l9: The abbreviations CPP and CPPS were introduced in the abstract, but not in the main text.
  • P1, l9: There is an ongoing debate about the usage of the terms CPP, CPPS and CPPPS. Please see the latest EAU Guidelines: Chronic Pelvic Pain for the distinction of these terms and clarify how you use these terms in context of your review.

Materials and Methods:

  • P2, ll89-90: Why did you search Medline in addition to searching the meta database PubMed, which uses, among other resources, Medline? Can the Virtual Health Library be regarded as a scientific database or could it be more likely classified as search engine?
  • P2, ll95-96: Why did you use these search term for the condition? Why did you not use other terms related to CPP?

Results:

  • Table 3: Please revise the columns Intervention group and Control group as it is not clear from the given information what treatment the participants received. Please avoid whole sentences like “They were given Terazosin once a day for 6 weeks.” Rather state the treatments and their parameters like duration, frequency, intensity, daily dose
  • Table 3: your legend does not fit to the content of the table
  • Table 4: desertion is a quite unusual word for dropouts; please consider rewording.
  • Table 4: Please revise the presentation of the results. It is confusing to read…. For example, what does that “The MPQ score showed great improvements in the intervention group” mean? Is it statistically significant and/or clinically relevant? What about the control group? Or what is the sense of presenting sentence like this: “Regarding the sphere of urinary symptoms both groups had improvements being greater in intervention group (p=0.7), compared to control group.”? Even if the symptoms seem to differ in their decrease among groups, there is no statistical difference between the groups… Perhaps your Table would benefit, if you just name the instrument and state the measurement values as well as the p values (intra- and intergroup comparisons).
  • P12, l201: Why did you include the two studies with a pain duration less than six months when you define CPP in your introduction as being present for at least six months?
  • P13, ll285-286: The abbreviations of the instruments were used before they were introduced later in the text.
  • The result section is relatively long, especially the section 3.6 Description of the variables and measurement tools. Please consider shortening.

Discussion/Conclusions:

  • P15, ll394-395: Why are the sample sizes a limitation? Did the authors reasoning their sample sizes, for example using sample size calculation? Why do you deem them to be (too) small?
  • P16, ll409-423: Please discuss your own methodology, your strengths and weaknesses instead of discussing limitations of the included studies.
  • P16, l420: Please discuss possible reasons why you were able to identify only one study with women despite the fact that more women suffer from chronic pelvic pain.
  • P16, l438: In the study from Brown et al., 13 out of 33 women dropped out, which is in contrast to your statement of low rates of abandonment. Please discuss the reasons of dropouts in the study and possible implications for the clinical practice.
  • P17, l459: Please put your conclusions into perspective and use “could be included” instead of “must be included”.
  • Please discuss the wording CPP, CPPS (and, if appropriate, CPPPS) and the implications for this review, for example identification of eligible studies.
  • Please discuss the controversies about magnetic field therapy, its evidence and effectiveness in general and in chronic pain entities in particular. Please also discuss the role of placebo in the included studies.

Author Response

ITEMIZED LIST OF THE REVIEWERS’ COMMENTS

Manuscript ID:  ijerph-1654750

Title: “Efficacy of magnetic therapy in pain reduction in patients with chronic pelvic pain: a systematic review”

Dear Reviewer,

We greatly appreciate the editor´s and reviewers’ kind and encouraging comments about our study. We have followed their suggestions, trying to incorporate them into the revised version of our manuscript. We uploaded the tracked changes manuscript, the clean version revised manuscript and itemized point-by-point response to the reviewer’ comments are presented below.

Editor´s and Reviewers´ comments:

*Reviewer 1

RV: Reviewer

AA: Authors

Abstract/Introduction:

RV: 1. P1, l9: The abbreviations CPP and CPPS were introduced in the abstract, but not in the main text.

AA: First, authors want to thank the modifications suggested by the reviewer and his/her effort to improve our manuscript. Following the reviewer´s recommendation, we have added the abbreviations CPP and CPPS in the main text in line 29 and 35 on page 1 in the revised clean version in the introduction section.

RV: 2. P1, l9: There is an ongoing debate about the usage of the terms CPP, CPPS and CPPPS. Please see the latest EAU Guidelines: Chronic Pelvic Pain for the distinction of these terms and clarify how you use these terms in context of your review.

AA: Following the reviewer´s recommendation, we have attempted to better clarify our introduction of the manuscript about the usage the terms CPP, CPPS and CPPPS. A new paragraph was added in lines 32 to 43 on page 1 in the revised clean version in the introduction section.

Materials and Methods:

RV: 3. P2, ll89-90: Why did you search Medline in addition to searching the meta database PubMed, which uses, among other resources, Medline? Can the Virtual Health Library be regarded as a scientific database or could it be more likely classified as search engine?

AA: Although these databases are interconnected, we consider that Medline, being a database specializing in biomedicine and biology, could be a resource that provides us with different types of information, since due to the profile of our article we wanted to make sure that we were not lose anything in the database given the paucity of existing literature on the subject.

On the other hand, it is true that the virtual health library, being a decentralized and dynamic collection of information sources, would be more appropriately classified as a search engine and not as a database. Then we have added it such as a search engine on lines 102-103 of page 3.

RV: 4. P2, ll95-96: Why did you use these search term for the condition? Why did you not use other terms related to CPP?

AA: These search terms were established because it was intended to cover the pathology in its entirety, since due to the lack of literature it was necessary not to limit the fields, in addition to being a novel subject whose pathophysiology and appropriate terminology were not well known. related to the pathology, it was decided to take a general approach and use the terms chronic pelvic pain (CPP) and chronic pelvic pain syndrome (CPPS), since both terms were frequently repeated in most of the articles found.

Results:

RV: 5. Table 3: Please revise the columns Intervention group and Control group as it is not clear from the given information what treatment the participants received. Please avoid whole sentences like “They were given Terazosin once a day for 6 weeks.” Rather state the treatments and their parameters like duration, frequency, intensity, daily dose

AA: Following the reviewer´s recommendation, we have improved the columns.

RV: 6. Table 3: your legend does not fit to the content of the table

AA: Sorry for the inconvenience. Following the reviewer suggestion, we have improved the legend.

RV: 7. Table 4: desertion is a quite unusual word for dropouts; please consider rewording.

AA: Sorry for the inconvenience. Following the reviewer suggestion, we have improved these words.

RV: 8. Table 4: Please revise the presentation of the results. It is confusing to read…. For example, what does that “The MPQ score showed great improvements in the intervention group” mean? Is it statistically significant and/or clinically relevant? What about the control group? Or what is the sense of presenting sentence like this: “Regarding the sphere of urinary symptoms both groups had improvements being greater in intervention group (p=0.7), compared to control group.”? Even if the symptoms seem to differ in their decrease among groups, there is no statistical difference between the groups… Perhaps your Table would benefit, if you just name the instrument and state the measurement values as well as the p values (intra- and intergroup comparisons).

AA: Sorry for the inconvenience. Following the reviewer suggestion, we have improved these words.

RV: 9. P12, l201: Why did you include the two studies with a pain duration less than six months when you define CPP in your introduction as being present for at least six months?

AA: Despite the fact that in our introduction it appeared (already modified) that chronic pelvic pain is from 6 months, in many articles in which we look for information it appears that this pathology is chronic from 3 months of duration, so Therefore, we see it convenient to have those patients who suffer from this pathology after 3 months of duration.

RV: 10. P13, ll285-286: The abbreviations of the instruments were used before they were introduced later in the text.

AA: Sorry for the inconvenience. Following the reviewer suggestion, we have improved this section.

RV: 11. The result section is relatively long, especially the section 3.6 Description of the variables and measurement tools. Please consider shortening.

AA: Sorry for the inconvenience. Following the reviewer suggestion, we have improved this section.

Discussion/Conclusions:

RV: 12.P15, ll394-395: Why are the sample sizes a limitation? Did the authors reasoning their sample sizes, for example using sample size calculation? Why do you deem them to be (too) small?

AA: We consider that it is a limitation because, for example, one article just had 21 participants, and another one just 33, and this can dead to dubious conclusions.

RV:13. P16, ll409-423: Please discuss your own methodology, your strengths and weaknesses instead of discussing limitations of the included studies.

AA: Following the reviewer´s recommendation, we have attempted to better clarify our discussion of our manuscript. Furthermore, a new paragraph was added in the first paragraph on page 9 in the revised clean version in the discussion section. Without a doubt, their recommendations help us to give more clarity and structure to the result and discussion part. Thank you very much for your comments.

RV:14. P16, l420: Please discuss possible reasons why you were able to identify only one study with women despite the fact that more women suffer from chronic pelvic pain.

AA: In the literature included in the review, the study samples were mostly men with chronic pelvic pain associated with chronic prostatitis. This fact of being able to limit the diagnosis to a specific organ provides more information about the possible treatments to be used in the foci Likewise, presenting a sample with greater homogeneity where it can be "categorically affirmed" that the pain originates in the prostate and not in other places allows for better guidance both in the diagnostic process and in treatment. On the other hand, in the article with the female sample there is greater heterogeneity due to the fact that there is a much more varied pathogenesis, this can give rise to numerous local syndromes with "organ specificity" but poorly defined, so that these multisystemic functional anomalies are difficult to define. require individual treatment, so it may be easier to carry out a study in men where the general aspects of pain are known, being able to extrapolate the treatment to patients with chronic prostatitis rather than in groups of women where limiting the diagnosis to a specific organ can overlook functional anomalies, which requires a more individualized treatment that is difficult to extrapolate to the rest of the female population with chronic pelvic pain.

RV:15. P16, l438: In the study from Brown et al., 13 out of 33 women dropped out, which is in contrast to your statement of low rates of abandonment. Please discuss the reasons of dropouts in the study and possible implications for the clinical practice.

AA: As explained by the researchers in the article by Brown et al, the original idea of ​​the study was to analyze the efficacy of magnetic field therapy through a 2-week double-blind study, and subsequently, those subjects who wanted, could complete optionally another two weeks. That is the reason for the dropouts, so only 19 participants wanted to continue in that second phase of the elective treatment.

RV:16. P17, l459: Please put your conclusions into perspective and use “could be included” instead of “must be included”.

AA: Totally agree with you. Following the reviewer´s recommendation, we have improved our conclusions. Thank you very much for your comments.

RV: 17. Please discuss the wording CPP, CPPS (and, if appropriate, CPPPS) and the implications for this review, for example identification of eligible studies.

AA: Following the reviewer´s recommendation, we have attempted to better clarify our discussion of the manuscript about the usage the terms CPP, CPPS and CPPPS. A new paragraph was added in lines 4 to 15 on page 1 in the revised clean version in the discussion section. Thank you very much for your comments.

RV: 18. Please discuss the controversies about magnetic field therapy, its evidence and effectiveness in general and in chronic pain entities in particular. Please also discuss the role of placebo in the included studies.

AA: Regarding the role of placebo, it is especially interesting that they used the same machine in several control groups but without turning it on, avoiding possible bias.

According to magnetic field therapy, this study has shown (in 4 of the 5 articles) that there is efficacy regarding pain in patients with CPP. In addition to pain, it has also been shown to improve quality of life and urinary symptoms.

Please, do not hesitate to contact me, if you require further corrections and information.

Thank you in advance

Reviewer 2 Report

Dear authors, 

I read with great interest the study entitled "Efficacy of magnetic therapy in pain reduction in patients with chronic pelvic pain: a systematic review". I found it an important topic of clinical importance based on the lack of gold standard therapy for the management of chronic pelvic pain which is a disease with pathogenesis that is still unknown while is related to significant morbidity for patients. However, there are severe methodological issues that preclude proceeding to publication. Some of them are listed below. 

Materials and methods

Page 2, lines 77-78: The authors should consider the revised PRISMA 2020 guidelines (please see PMID 33782057)

Page 2, lines 85-86: Probably the authors should also add here the pilot studies.

Page 4, lines 114-119: I think that the initials of the 3rd reviewer should also be listed above along with the reference on the total of 3 reviewers that finally evaluated the eligible studies.

Page 5, line 126: What about the quality assessment of the pilot studies?

Results 

The whole 3.5 section is too wordy and represents a repetition of the information provided by the included studies and not a critical appraisal of them. The authors should reduce and critically revise the section. Some of the recorded information should be a part of tables.

In addition some of the information should be a part of materials and methods of the study. The results section should summarise and combine the main outcomes reported by the included studies (as is done in subsection 3.6) and not the methods used by each study. As a result the whole section needs extensive revision.

Minor

Lines 208-209: As far as I am concerned the pathophysiology of CPP is totally different among men and women. Therefore, a separate analysis should be beneficial.

Discussion

The section is poorly analysed and lacks main information that must be included in a discussion section. Please look again at PRISMA guidelines and revise accordingly.

Author Response

ITEMIZED LIST OF THE REVIEWERS’ COMMENTS

Manuscript ID:  ijerph-1654750

Title: “Efficacy of magnetic therapy in pain reduction in patients with chronic pelvic pain: a systematic review”

Dear Reviewer,

We greatly appreciate the editor´s and reviewers’ kind and encouraging comments about our study. We have followed their suggestions, trying to incorporate them into the revised version of our manuscript. We uploaded the tracked changes manuscript, the clean version revised manuscript and itemized point-by-point response to the reviewer’ comments are presented below.

Editor´s and Reviewers´ comments:

*Reviewer 2

RV: Reviewer

AA: Authors

General

Dear authors, 

I read with great interest the study entitled "Efficacy of magnetic therapy in pain reduction in patients with chronic pelvic pain: a systematic review". I found it an important topic of clinical importance based on the lack of gold standard therapy for the management of chronic pelvic pain which is a disease with pathogenesis that is still unknown while is related to significant morbidity for patients. However, there are severe methodological issues that preclude proceeding to publication. Some of them are listed below. 

Materials and methods

RV: 1. Page 2, lines 77-78: The authors should consider the revised PRISMA 2020 guidelines (please see PMID 33782057)

AA: Thank you for your comments. We totally agree with you. Following the reviewer’s recommendation, we have modified the most recent reference about PRISMA in the revised clean version.

RV: 2. Page 2, lines 85-86: Probably the authors should also add here the pilot studies.

AA: Thank you for your suggestion. We agree with your comment. We have added “the pilot studies”.

RV: 3. Page 4, lines 114-119: I think that the initials of the 3rd reviewer should also be listed above along with the reference on the total of 3 reviewers that finally evaluated the eligible studies.

AA: We agree with you. Following the reviewer’s recommendation, we have added the initials of the 3rd reviewer where you advised us. Thank you for your comments.

RV: 4. Page 5, line 126: What about the quality assessment of the pilot studies?

AA: Both pilot studies had an evidence level of 1 and good quality.

Results 

RV: 5. The whole 3.5 section is too wordy and represents a repetition of the information provided by the included studies and not a critical appraisal of them. The authors should reduce and critically revise the section. Some of the recorded information should be a part of tables.

AA: Thank you for your suggestion. We agree with your comment. We have shortened this section.

RV: 6. In addition some of the information should be a part of materials and methods of the study. The results section should summarise and combine the main outcomes reported by the included studies (as is done in subsection 3.6) and not the methods used by each study. As a result the whole section needs extensive revision.

AA: Thank you for the advice, we have removed the text as you requested.

Discussion

RV: 7. The section is poorly analysed and lacks main information that must be included in a discussion section. Please look again at PRISMA guidelines and revise accordingly.

AA: Following the reviewer´s recommendation, we have attempted to better clarify our discussion. In addition to this, we have added the usage the terms CPP, CPPS and CPPPS. Please let me know if you require any further changes.

Please, do not hesitate to contact me, if you require further corrections and information.

Thank you in advance

Reviewer 3 Report

thank you to giving me the chance to review this quantitative review on efficacy of magnetic therapy in pain reduction in patients with chronic pelvic pain.

The authors correctly made literature review, data extraction, analyses and interpretation. The authors extensively discussed about strenght and limitations of their findings (few studies, scanty data on women, etc)

I suggest to include some details regarding the pelvic floor dysfunction in patients with chronic pelvic pain (doi: https://doi.org/10.1002/ijgo.14088; 10.1007/s11916-012-0277-8 )

Author Response

ITEMIZED LIST OF THE REVIEWERS’ COMMENTS

Manuscript ID:  ijerph-1654750

Title: “Efficacy of magnetic therapy in pain reduction in patients with chronic pelvic pain: a systematic review”

Dear Reviewer,

We greatly appreciate the editor´s and reviewers’ kind and encouraging comments about our study. We have followed their suggestions, trying to incorporate them into the revised version of our manuscript. We uploaded the tracked changes manuscript, the clean version revised manuscript and itemized point-by-point response to the reviewer’ comments are presented below.

Editor´s and Reviewers´ comments:

*Reviewer 3

RV: Reviewer

AA: Authors

Thank you to giving me the chance to review this quantitative review on efficacy of magnetic therapy in pain reduction in patients with chronic pelvic pain.

The authors correctly made literature review, data extraction, analyses and interpretation. The authors extensively discussed about strenght and limitations of their findings (few studies, scanty data on women, etc)

RV: 1. I suggest to include some details regarding the pelvic floor dysfunction in patients with chronic pelvic pain (doi: https://doi.org/10.1002/ijgo.14088; 10.1007/s11916-012-0277-8 )

AA: First, authors want to thank the modifications suggested by the reviewer and his/her effort to improve our manuscript. Following the reviewer´s recommendation, we have attempted to better clarify regarding the pelvic floor dysfunction in patients with chronic pelvic pain. A new paragraph was added in lines 49 to 54 on page 2 in the revised clean version in the introduction section in the revised clean version. We have added the following two references:

*Raimondo D, Cocchi L, Raffone A, Del Forno S, Iodice R, Maletta M, Aru AC, Salucci P, Ambrosio M, Mollo A, Youssef A, Casadio P, Seracchioli R. Pelvic floor dysfunction at transperineal ultrasound and chronic constipation in women with endometriosis. Int J Gynaecol Obstet. 2022 Jan 7. doi: 10.1002/ijgo.14088. Epub ahead of print. PMID: 34995357.

*Kotarinos R. Myofascial pelvic pain. Curr Pain Headache Rep. 2012 Oct;16(5):433-8. doi: 10.1007/s11916-012-0277-8. PMID: 22648177.

Please, do not hesitate to contact me, if you require further corrections and information.

Thank you in advance

All the best

Reviewer 4 Report

Firstly, I would like to congratulate the authors on a very well-conducted review. The study has merit and will be of interest to our readers. However, there are certain issues that need to be highlighted. The manuscript needs revision before it can be reconsidered.

Introduction: well-written.

-Please add your hypothesis in 2-3 lines at the end of the Introduction section.

Methods: well-written

-Why have you written/searched Medline and PubMed separately? PubMed is the database that provides access to Medline. However, the PubMed database also provides access to PubMed Central, etc.

Please clear this issue. If you have searched Medline separately, please highlight which articles were found in PubMed but not in Medline.

-In cases of scoring systems for methodological quality assessments, it is always better to engage two observers to perform independent scoring of the included studies. Subsequently, an inter-observer agreement can be performed. What was performed in the present review?

Results:

-Sections 3.5 and 3.6 are very detailed. They can be trimmed as there is a lot of redundant information.

-Please do not duplicate the information in the text that is already provided in the tables. There is a lot of information that is given in Tables 3 and 4 and also duplicated in sections 3.4, 3.5, and 3.6. Please delete it.

Discussion:

-Well-written. No changes are needed.

Author Response

ITEMIZED LIST OF THE REVIEWERS’ COMMENTS

Manuscript ID:  ijerph-1654750

Title: “Efficacy of magnetic therapy in pain reduction in patients with chronic pelvic pain: a systematic review”

Dear Reviewer,

We greatly appreciate the editor´s and reviewers’ kind and encouraging comments about our study. We have followed their suggestions, trying to incorporate them into the revised version of our manuscript. We uploaded the tracked changes manuscript, the clean version revised manuscript and itemized point-by-point response to the reviewer’ comments are presented below.

Editor´s and Reviewers´ comments:

*Reviewer 4

RV: Reviewer

AA: Authors

General

Firstly, I would like to congratulate the authors on a very well-conducted review. The study has merit and will be of interest to our readers. However, there are certain issues that need to be highlighted. The manuscript needs revision before it can be reconsidered.

Introduction: well-written.

RV: 1. Please add your hypothesis in 2-3 lines at the end of the Introduction section.

AA: Following the reviewer´s recommendation, we have added the hypothesis in the main text on page 2 in the revised clean version in the introduction section.

The new sentence in the introduction section now reads:

The hypothesis was that magnetic field therapy should have pain reduction and improved quality of life in men and women with CPP”. 

Methods: well-written

RV: 2. Why have you written/searched Medline and PubMed separately? PubMed is the database that provides access to Medline. However, the PubMed database also provides access to PubMed Central, etc.

Please clear this issue. If you have searched Medline separately, please highlight which articles were found in PubMed but not in Medline.

AA: Although these databases are interconnected, we consider that Medline, being a database specializing in biomedicine and biology, could be a resource that provides us with different types of information, since due to the profile of our article we wanted to make sure that we were not lose anything in the database given the paucity of existing literature on the subject. In fact, we obtained articles that do not appear in Pubmed, although they were later excluded because they did not answer our PICO question.

The articles present in medline that did not appear in pubmed were the following:

  • Anorectal Disorders: An Update.
  • Cortical neurostimulation for neuropathic pain: state of the art and perspectives.
  • Patient-reported outcomes and outcome measures in childbirth perineal trauma research: a systematic review.
  • A glimpse into the efficacy of alternative therapies in the management of benign prostatic hyperplasia.
  • The effect of biofeedback interventions on pain, overall symptoms, qyuality of life and physiological parameters in patients with pelvic pain: a systematic review.

RV: 3. In cases of scoring systems for methodological quality assessments, it is always better to engage two observers to perform independent scoring of the included studies. Subsequently, an inter-observer agreement can be performed. What was performed in the present review?

AA: “Three reviewers (ADP, MJR and RMV) assessed the methological quality. In case of doubt, authors resolved disagreements by consensus and consulting a fourth author (EDM) when necessary”. We have added this clarification on the page 5 in the revised clean version.

Results:

RV: 4. Sections 3.5 and 3.6 are very detailed. They can be trimmed as there is a lot of redundant information.

AA: Thank you for the advice, we have shortened the text as you requested.

RV: 5. Please do not duplicate the information in the text that is already provided in the tables. There is a lot of information that is given in Tables 3 and 4 and also duplicated in sections 3.4, 3.5, and 3.6. Please delete it.

AA: Thank you for the advice, we have removed the text as you requested.

Discussion:

-Well-written. No changes are needed.

AA: Without a doubt, their recommendations help us to give more clarity and structure to the result part. Thank you very much for your comments.

Please, do not hesitate to contact me, if you require further corrections and information.

Thank you in advance

Kind Regards

Round 2

Reviewer 2 Report

Dear authors, 

a significant progress has been done by the authors for the improvement of the article entitled "Efficacy of magnetic therapy in pain reduction in patients with chronic pelvic pain: a systematic review". However, my main concerns on the inappropriate analysis of discussion still remain unchanged. A significant part of the discussion still analyses the outcomes of the study. More specifically, no change has been detected among the initial and the revised version of the discussion. 

Author Response

ITEMIZED LIST OF THE REVIEWERS’ COMMENTS

Manuscript ID:  ijerph-1654750

Title: “Efficacy of magnetic therapy in pain reduction in patients with chronic pelvic pain: a systematic review”

Dear Reviewer,

We greatly appreciate the editor´s and reviewers’ kind and encouraging comments about our study. We have followed their suggestions, trying to incorporate them into the revised version of our manuscript. We uploaded the tracked changes manuscript, the clean version revised manuscript and itemized point-by-point response to the reviewer’ comments are presented below.

Editor´s and Reviewers´ comments:

*Reviewer 2

RV: Reviewer

AA: Authors

RV: Dear authors, 

A significant progress has been done by the authors for the improvement of the article entitled "Efficacy of magnetic therapy in pain reduction in patients with chronic pelvic pain: a systematic review". However, my main concerns on the inappropriate analysis of discussion still remain unchanged. A significant part of the discussion still analyses the outcomes of the study. More specifically, no change has been detected among the initial and the revised version of the discussion. 

AA: First, authors want to thank the modifications suggested by the reviewer and his/her effort to improve our manuscript. Following the reviewer´s recommendation, we have attempted to better clarify our discussion of the manuscript. Without a doubt, their recommendations help us to give more clarity and structure our limitations part. We agree with your comment. We have shortened limitations section. A new paragraph about limitation was added added in lines 360 to 369 on page 14 in the revised clean version in the discussion section. Thank you very much for your suggestion.

The new paragraph in the limitation section now reads:

There are several limitations to the analysis that should be considered when interpreting these results. First, heterogeneity among the different studies was so extensive that a meta-analysis could not be performed. It was difficult to find articles about the study topic, despite the search conducted in multiple electronic databases of published studies.  Second, there was little uniformity in study populations, the small sample sizes, interventions and their duration, measured variables, and different measurement instruments. Because all the articles presented in this review are quite heterogeneous in certain parameters; moreover, one of them presents moderate methodological quality [21]. Therefore, the results must be interpreted with caution, since the studies included in the review used different devices, treatment time and time between sessions”

Please, do not hesitate to contact me, if you require further corrections and information.

Thank you in advance

Kind Regards,

Reviewer 4 Report

In the revised manuscript, the authors have incorporated all my comments. The overall scientific quality of the manuscript has improved significantly. I congratulate the authors for their work.

Author Response

ITEMIZED LIST OF THE REVIEWERS’ COMMENTS

Manuscript ID:  ijerph-1654750

Title: “Efficacy of magnetic therapy in pain reduction in patients with chronic pelvic pain: a systematic review”

Dear Reviewer,

We greatly appreciate the editor´s and reviewers’ kind and encouraging comments about our study. We have followed their suggestions, trying to incorporate them into the revised version of our manuscript. We uploaded the tracked changes manuscript, the clean version revised manuscript and itemized point-by-point response to the reviewer’ comments are presented below.

Editor´s and Reviewers´ comments:

*Reviewer 4

RV: Reviewer

AA: Authors

RV: In the revised manuscript, the authors have incorporated all my comments. The overall scientific quality of the manuscript has improved significantly. I congratulate the authors for their work.

AA: First, authors want to thank the modifications suggested by the reviewer and his/her effort to improve our manuscript. Thank you very much for your comments.

Kind Regards,
